# Self-Compacting Alkali-Activated Materials: Progress and Perspectives

**DOI:** 10.3390/molecules27010081

**Published:** 2021-12-23

**Authors:** Zengqing Sun, Qingyu Tang, Xiaohui Fan, Min Gan, Xuling Chen, Zhiyun Ji, Xiaoxian Huang

**Affiliations:** School of Minerals Processing and Bioengineering, Central South University, Changsha 410083, China

**Keywords:** self-compacting, alkali-activated materials, workability, mechanical strength, durability

## Abstract

Alkali-activated materials (AAMs) are considered to be alternative cementitious materials for civil infrastructures. Nowadays, efforts have been made in developing AAMs with self-compacting ability. The obtained self-compacting AAMs (SCAAMs) accomplish superior passing and filling properties as well as excellent mechanical and environmental advantages. This work critically revisits recent progresses in SCAAMs including mixture proportions, fresh properties, mechanical strength, microstructure, acid and sulfate resistance, high temperature behaviors, impact resistance and interface shear strength. To facilitate direct comparison and interpretation of data from different publications, mixture proportions were normalized in terms of the content of key reactive components from precursors and activators, and correlation with mechanical behaviors was made. Moreover, special attention was paid to current research challenges and perspectives to promote further investigation and field application of SCAAMs as advanced construction material.

## 1. Introduction

Self-compacting concrete (SCC) is a kind of concrete that can be placed without any mechanical consolidation while keeping stable composition [1,2]. The history of SCC can be dated back to 1986, when Okamura first proposed the concept [3], and in 1988, when Ozawa from the University of Tokyo developed the prototype [4]. Extensive investigations have since been conducted on the production and placement of SCC, as well as the characterization of fresh and hardened properties [1,2,5,6]. In general, SCC possess advantages such as eliminated vibration need, improved filling capacity, decreased permeability, enhanced durability, reduced construction time and labor cost, etc. [7,8,9,10,11]. Thus, SCC can be used particularly in pre-cast, high rise buildings requiring congested reinforcement.

The construction industry is now seeking alternative cementitious materials, partially because the production of cement clinker is associated with huge amounts of greenhouse gas emissions. It is estimated that cement production alone accounts for approximately 8% of global overall CO_2_ emission [12,13,14,15,16]. Alkali-activated materials (AAMs) are among the more robust candidates, with excellent engineering properties and reduced CO_2_ emission. Solid precursors used for AAM production are normally industrial and agricultural by-products/wastes, such as ground granulated blast furnace slag (GGBS), fly ash (FA), coal gangue, red mud, rice husk ash (RHA), etc. Depending on the reaction mechanism stemming from initial mixture proportion, AAMs are generally divided into high-calcium systems (with the representative of alkali activated GGBS (AAS)) and low-calcium systems (also known as geopolymer) [17]. The differences in reaction processes and products have been well investigated and documented [17,18]. Studies have reported that AAMs possess properties including high mechanical strength, thermal and chemical resistance, strong adhesion to different surfaces, etc. Hence, AAMs have been successfully utilized in the production of masonry blocks, retail buildings, storehouses, drainage systems, and even airport pavement since the 1970s [18,19,20].

Nowadays, developments have also been made in self-compacting AAMs (SCAAMs), which combines the advantages of superior passing and filling properties of SCC with the excellent mechanical and environmental properties of AAMs. Memon et al. synthesized SCAAMs using FA as solid source and characterized the influence of activator concentration, liquid-solid ratio and curing temperature [21,22,23,24,25]. Manjunath and Narasimhan investigated the flowability, passing ability, compressive strength, and water absorption of slag-based SCAAM [26]. Based on the findings, mixtures with high performances, i.e., compressive strength up to 90 MPa and water absorption of 2.1–2.7%, were produced [27]. While in [9,28,29,30], RHA, nano silica (NS), ceramic powder (CP) and Wollastonite were independently utilized as reactive additive. All these facilitate the investigation of SCAAMs synthesis and properties, but also make it difficult to compare the obtained results and figure out the dominating parameters.

In this review, the synthesis of SCAAMs based on different precursors and mixture proportions was outlined. For comparison and discussion, mixtures were recalculated and normalized in terms of the content of key reactive components from precursors and activators. Workability, mechanical properties, microstructure, acid and sulfate resistance, high temperature behaviors, impact resistance and interface shear strength were addressed. Special attention is paid to current research challenges and perspectives to boost further investigation and promote field application of SCAAMs.

## 2. Materials and Formulations

GGBS and FA are among the most frequently used solid precursors for AAM production including SCAAMs. The physical and chemical characteristics of these materials have been well characterized and described in literature [17,31,32]. Recently, studies have been conducted using other materials as solid sources for SCAAMs preparation. These solid materials provide desirable properties, minimize the total environmental footprint, and widen the sources and types of precursors, which are briefly described in this section.

### 2.1. New Precursors

#### 2.1.1. Rice Husk Ash

RHA is a typical and plentifully accessible agricultural waste generated from rice milling plants during the burning of husk under controlled temperature. According to a report from the Food and Agricultural Organization (FAO), 741.3 million tons of RHA was produced all over the world [28]. RHA is predominantly composed of SiO_2_, which can be 80–95% of its total. CaO, MgO, K_2_O, Al_2_O_3_, Fe_2_O_3_ and Na_2_O are frequently detected in RHA but in minor quantities [33,34]. Efforts have been made in utilizing RHA as SCM for concrete production or as a precursor for AAMs synthesis [35]. Generally, the inclusion of RHA contributed to a refined pore structure and generated a dense microstructure, which consequently resulted in the enhancement of mechanical properties [36,37,38,39,40,41,42,43,44,45]. The influence of RHA on fresh and hardened properties of GGBS-based SCAAM has been investigated [28]. The flowability and fluidity reduced as RHA incorporation increased from 5% to 25%, but the workability of the tested mixtures remained within acceptable limits. Mechanical improvement was only observed at RHA replacement of 5%. Similar results were reported by Ardiantoro et al. [46], in which the mixture of RHA and fly ash was used as precursor and maximum substitution of RHA was 8.65% in terms of strength behavior. More detailed investigation is needed to optimize and promote the utilization of RHA in SCAAM.

#### 2.1.2. Nano Silica

Nano silica (NS), also known as silica nanoparticles or silicon dioxide nanoparticles, is an excellent additive for the manufacture of plastics, rubber, catalyst and coating materials [47]. NS has also been utilized as effective supplementary cementitious material (SCM) in Portland cement system. NS possesses ideal pozzolanic reactivity, which can be assigned to the X-ray amorphous characteristic as well as the considerable fine particle size. Acceleration of hydration process and refinement of pore structure have been observed after adding NS, which consequently enhanced the mechanical properties of the obtained concrete [48,49,50,51,52]. Moreover, the inclusion of NS in small amounts can obviously decrease setting time and improve the mechanical strength of AAM [53,54]. Increased formation of reaction products and densified matrix structures were observed, which also contributed to improved durability properties reflecting less water absorption and reduced charge pass in a repaid Cl^−^ penetration test [54]. SCAAM based on alkali activated FA-GGBS with varying NS content has been investigated [28]. The inclusion of NS by 1% and 2% decreased slump flow of SCAAM from 709 mm to 695 mm and 680 mm, respectively.

#### 2.1.3. Ceramic Powder

Ceramic powder (CP) is obtained after the collection, cleaning and grinding of waste ceramic, which is mainly composed of SiO_2_, Al_2_O_3_ and Na_2_O, with main crystalline phases of quartz and mullite. [55,56,57,58]. Due to the low pozzolanic reactivity and high water adsorption property, the utilization of CP in ordinary Portland cement (OPC) concrete production as SCM normally leads to reduced strength [57,59]. Meanwhile, geopolymer has been synthesized by Sun et al. [60] using CP as precursor, with obtained geopolymer showing excellent mechanical strength and thermal resistance. The mixture of CP and GGBS (CP accounts up to 80 wt.%) has been used to prepare SCAAM [9], with the rheological and mechanical properties of the resulting SCAAM being characterized. The addition of CP contributes to enhance the flow and passing ability of SCAAM but causes a decrease in segregation resistance. Though the strength decreased with the increase of CP, the tested samples achieved acceptable compressive strength.

#### 2.1.4. Wollastonite

Wollastonite, a calcium inosilicate mineral, is composed of calcium and silicon oxides with small substitutions, if any, of the calcium by iron, magnesium and manganese. Geologically, it is usually formed by placing calcium rock (for example limestone, dolomite) under high temperature and pressure with the presence of fluids containing silica. Wollastonite is characterized by a unique needle-like structure, high whiteness/brightness and thermal stability, low moisture, and oil absorption [61]. Wollastonite has been used primarily in the production of ceramics and brakes, or as fillers for plastics, paints, etc. [62]. Moreover, several studies have investigated the characteristics of concrete incorporating wollastonite. Enhancements in compressive strength have been reported by Ransinchung and Kumar [63] when replacing cement by wollastonite up to 15%. Higher replacement caused a decrease in both compressive and flexural strength as well as marginal variations in pull-off tests [64]. In contrast, the inclusion of wollastonite accelerated the dissolution of solid precursor and consequently contributed to increase viscosity and mechanical strength of obtained AAM [65]. SCAAM utilizing the mixture of FA, GGBS and wollastonite as precursor was prepared by Vishnu et al. [27]. Comparing with the control specimen, the synthesized SCAAM containing 10% wollastonite achieved better workability and similar strength.

#### 2.1.5. Water Treatment Sludge

Water treatment sludge (WTS) is the solid waste generated from water treatment plants, mainly during the coagulation-flocculation step. In many cases, WTS can be defined as clay-based waste composing suspended aluminosilicate-rich sediments and by-products of chemical reagents [66]. WTS has been previously disposed of by landfilling, while nowadays efforts have been made to convert WTS into alternative source materials for ceramic and cement production [67,68]. In the regard of AAM production, different synthesis processes have been employed. Guo and Shi [69], Nimwinya et al. [70], and Waijarean et al. [71], individually synthesized AAM using WTS after calcination. The thermal treatment process not only removes organic matter present in WTS but also enhances pozzolanic reactivity of the resultant sludge. The 28 d compressive strength over 50 MPa has been achieved by AAM produced using calcined WTS [69]. WTS has been used as solid precursor for AAM synthesis without thermal treatment [72,73,74]. The incorporation of non-calcined WTS may generally induce undesirable characteristics such as delayed setting time, reduced workability, and decreased mechanical strength [72]. For modification, WTS was mixed with other pozzolanic precursors or curing AAM specimens at elevated temperatures [72,74,75]. Hwang et al. [75] characterized the fresh and hardened properties of SCAAM containing up to 20% WTS. The authors reported that increasing WTS content led to reduced workability, enhanced fresh unit weight, and increased initial setting time of fresh SCAAM. Moreover, the compressive strength of hardened samples was significantly enhanced by the increase of WTS.

### 2.2. Formulations of SCAAMs

The mixture SCAAM is designed to achieve required performances in both fresh and hardened states. Though lack of design code, some of the approaches have been demonstrated as applicable in the fabrication of SCAAM. Table 1 summarizes typical SCAAM mixtures from the literature. The formulations of solid precursor and activator possess dominant roles on the workability of AAM, which will be discussed in depth in the following section. Due to the extremely alkaline conditions, subsequent degradation of superplasticizer in AAM system has been observed [76]. Hence, many effective superplasticizers from the OPC system do not work well for AAM. Meanwhile, polycarboxylate-based superplasticizers have been demonstrated in some studies showing promising performance in AAM [77,78,79]. The polycarboxylate-based superplasticizer has also been adopted [30,79,80]. Success in workability improvement has been achieved after adding superplasticizer, but the addition amount is much higher than in cement concrete. Further investigation to minimize the required quantity or to explore other effective chemical admixtures are needed.

SCAAM is normally of higher paste to aggregate and fine aggregate to coarse aggregate ratios. The adding sequence of components, mixing time and even mixer type affect the homogeneity and uniformity of obtained mixture. Based on the literature cited in this work, fine and coarse aggregates were firstly mixed followed by the addition of solid precursor. Activator, superplasticizer, and extra water (if any) were then added. The whole mixing time ranges from 6 to 9 min or even longer, depending on initial mix design, required property and mixer. Afterwards, the homogeneous mixture was subjected to fresh state characterization or casted and cured for hardened property tests.

## 3. Property of Fresh SCAAMs

Fresh properties of typical SCAAMs from the literature are listed in Table 1. It should be mentioned that GGBS, FA and other precursors/additives of different physicochemical characteristics were used. The mixture proportions were also described in various ways, including activator/solid ratio, Na_2_O/binder ratio, activator ratio, etc. To facilitate and simplify the comparison of data from different sources, recalculation was conducted with the following principles:

(1) alkaline solution is composed of solid activator plus water, i.e., in sodium silicate solution is composed of SiO_2_ + Na_2_O + H_2_O, NaOH solution is composed of Na_2_O + H_2_O;

(2) binder is the sum of precursor and solid activator and additives (RHA, NS, etc., if any);

(3) total water is the sum of water from alkaline solution and extra water added. 

For example, an SCAAM is synthesized using the mixture of GGBS (SiO_2_: 30.8 wt.%) and FA (SiO_2_: 57.2 wt.%), the content of precursor mixture is 484 kg/m^3^, with mass ratio of GGBS to FA being 7:3. NaOH solution with concentration of 2 mol/L and analytical sodium silicate solution (Modulus: 2.07, SiO_2_: 29.5 wt.%, Na_2_O: 14.70 wt.%, H_2_O: 55.80 wt.%) were used. Mass ratio of sodium silicate to NaOH solution was 0.75, mixed solution was cooled down for 24 h before being used. For the concrete preparation, river sand and crushed gravel were used. The solution-to-solid ratio, extra water, fine and coarse aggregate content was 0.5, 185.9 kg/m^3^, 844 kg/m^3^ and 756 kg/m^3^, respectively.

In this case, for 1 m^3^ of SCAAM concrete,

mass of mixed solution: m_solution_ = 484 kg/m^3^ × 0.5 × 1m^3^ = 242 kg;

mass of solidum silicate solution: m_silicate_ = 242 kg × 0.429 = 103.71 kg;

mass of NaOH solution: m_NaOH_ = 242 kg × 0.571 = 138.29 kg;

mass of SiO_2_ in sodium silicate solution: m_(SiO2-solution)_ = 103.71 kg × 0.295 = 30.60 kg;

mass of SiO_2_ in GGBS: m_(SiO2-GGBS)_ = 484 kg × 0.× 0.308 = 104.35 kg;

mass of SiO_2_ in FA: m_(SiO2-FA)_ = 484 kg × 0.3 × 0.572 = 83.05 kg;

mass of Na_2_O in sodium silicate solution: m_(Na2O-SS)_ = 103.71 kg × 0.147 = 15.25 kg;

mass of Na_2_O in NaOH solution: m_(Na2O-SH)_ = (138.29 kg × 0.74 × 62)/(40 × 2) = 79.31 kg;

mass of binder: m_binder_ = 484 kg + 30.60 kg + 15.25 kg + 79.31 kg = 609.16 kg;

mass of total water: m_water_ = 103.71 kg × 0.558 + 138.29 kg − 79.31 kg + 185.9 kg = 3 02.75 kg;

water-to-binder ratio: r = m_water_/m_binder_ =(302.75 kg)/(609.16 kg) = 0.497;

content of SiO_2_ in binder: w_SiO2_ = (m_(SiO2-solution)_ + m_(SiO2-GGBS)_ + m_(SiO2-FA_))/m_binder_ × 100% = 35.79%.

After this kind of calculation, the mixtures from different literature can be directly compared and the relationship between slump flow value and mixture proportion is plotted in Figure 1.

As abovementioned, water content is one of the key factors determining the fresh properties of SCAAM. In [86], the slump flow of SCAAM increased close to linearly from 650 mm to 825 mm as the water/binder ratio increased from 0.4 to 0.5, which all exceeded the spread values as per ENFARC and the Australia code for SCC [87]. The water/binder ratio is located in a similar range to conventional AAMs. The much higher slump range can be related to the proportional binder design and the aggregate gradation. SCC normally has a higher paste-to-aggregate ratio than conventional concrete, and the volumetric content of coarse aggregate is less than 50% [2,6]. Alkaline solution initially contains a considerable amount of water, and enhancing the solution/solid ratio can, as a consequence, increase the total water content. Meanwhile, increased activator dosage plays a significant role in accelerating setting and hardening of resultant SCAAMs. Thus, alkaline activator dosage has fluctuating effects on the workability of SCAAMs, and in some cases reduces workability [88]. As per the concentration of NaOH solution, a drop in workability induced by concentration increase has been reported [88,89]. Saini and Vattipalli [89] observed workability decrease in GGBS-based SCAAM when the NaOH concentration varied between 10 mol/L and 16 mol/L. Nagaraj and Venkatesh Babu stated that mixtures with NaOH concentration above 12 mol/L failed to meet the workability requirements for SCC [88].

It can be inferred from the published results (shown in Figure 1) that workability improves with the replacement of GGBS by FA [81,84,85]. Huseien and Shah [85] reported that replacing 70% GGBS by FA resulted in slump increase of 70 mm. The slump flow value of 100% FA-based SCAAM has been reported as 1.25 and 1.03 times its counterparts with 50% GGBS and 100% GGBS, respectively [81]. A similar conclusion can also be drawn for V-funnel, L-box, and J-ring tests [84]. GGBS is generally of higher reactivity than FA, which might correspond to momentous acceleration in alkaline activation, hardening and setting. In addition, unlike the angular shaped particles of GGBS, the spherical particle of FA also contributes to improve workability [81].

## 4. Property of Hardened SCAAMs

### 4.1. Compressive Strength

Reactivity of solid precursors has a significant influence on the mechanical properties of resultant SCAAMs. Specimens prepared using FA were impossible to demold after 3 d curing at ambient condition because of low reactivity [28]. In contrast, GGBS-based SCAAM achieved 3 d compressive strength of 34.76 MPa, accounting for 84% of 28 d strength [28]. Studies show that the activation energy of the FA system is 1.5 times that of the GGBS system or even higher [31,90,91], indicating that the FA system must overcome a higher energy barrier than GGBS system to initiate reaction. Solutions to lower the activation energy include proportional mix-design and elevated temperature curing. Combining FA with GGBS is a powerful way to improve the mechanical behaviors of SCAAMs containing FA. According to Nagaraj and Venkatesh Babu [81], replacing 50% of FA by GGBS can individually enhance the 7 d and 90 d compressive strength by 3.34–5.58 times and 3.64–5.23 times, respectively. Apart from the improved reaction kinetic after incorporating GGBS, the modification of microstructure was also observed, which will be discussed in Section 5.2 [81]. Another way in terms of mix-design is modifying activator utilization. It has been clearly observed that the strength of SCAAMs increased with an increase in activator molarity [88]. Low strength has been attained by samples prepared with solidum hydroxide solution lower than 4 M. A high activator-to-solid ratio would also have some negative influence on strength [88,92]. As discussed above, considerable content of water was incorporated into SCAAM matrixes of high activator-to-solid ratio, which would lead to a porosity increase and mechanical deterioration.

Patel and Shah [28] have stated that the dissolution of Si and Al in FA particles is slow at ambient temperature. Curing at elevated temperature is of a benefit for the formation of gels and the following polymerization and condensation processes. The mechanism behind this phenomenon has been investigated by Sun and Vollpracht [31]. The alkali-activated FA system is more temperature dependent, as raising the reaction temperature can significantly enhance cumulative isothermal heat release. Moreover, the calorimetric response was converted from single-peak into multiple-peak, indicating the occurrence of multiple chemical reactions and structure transformation [31]. Supporting the statement of Patel and Shah [28], the 3 d compressive strength of obtained samples can be increased 8.8-fold after elevating the first 12 h curing time from 20 °C to 60 °C, while extending duration from 12 h to 24 h can further improve the mechanical performance 1.5-fold [93].

It should be mentioned that data from the literature can hardly be directly compared since mix design and synthesis procedure differ widely. After revisiting the literature, a calculation was conducted to normalize the binder composition. The results are shown in Figure 2, which depicts the pseudo-ternary plot relating the effects of the main components to the 28 d compressive strength of resultant SCAAMs. The calculation was based on the same principles as Section 5. In particular, the Na_2_Oeq is calculated following:Na_2_O_eq_ = (m(Na_2_O) + m(K_2_O)/94.2 × 62 + n(CaO)/56.1 × 62)/mbinder × 100%
in which the contents of K_2_O and CaO are included on the basis of molar equivalency and cation charge. The authors are highly aware that this kind of calculation may underestimate the influence of CaO, especially for SCAAMs with high calcium. The role of calcium differs from sodium/potassium in alkali activation process. As the low-calcium is predominantly composed of alkali aluminosilicate gel (N-A-S-(H)), in which the alkali cations balance the negative charge caused by substitution of silicon by aluminum [17]. Meanwhile, the calcium aluminosilicate gel (C-A-S-H) is detected as the key reaction product, which is to some extent regarded as aluminum substituted calcium silicate gel (C-S-H). The C-A-S-H gel is reported to possess a microstructure similar to tobermorite, with calcium cations locating in the interlayer region of tetrahedrally coordinated silicate chains [17]. But this calculation can somehow facilitate the comparison of independent studies and afford insights into the contribution of main components to mechanical performance of obtained products.

It can then be inferred that the increase of Na_2_Oeq in binder generally leads to an improvement of compressive strength. Multi-linear regression has been made for 28 d compressive strength and Na_2_O dosage was found as the key positive parameter in the correlation [27]. High Na_2_Oeq content suggests high alkalinity after mixing with water, which consequently contributes to the dissolution of aluminosilicate components and the formation of strength-giving products. 

It has been stated that AAMs can be produced with a Si/Al ratio ranging from 0.5 to 300 [39]. Xu and Deventer [95] observed that a correlation is present between the Si/Al ratio in mixture and mechanical strength of AAMs. Thus, a feasible Si-Al proportion is required to obtain materials with interesting properties. Based on the chemistry of zeolite, Davidovits recommended a Si/Al ratio of 1.75–2.25 (SiO_2_/Al_2_O_3_ ratio 3.5–4.5) for metakaolin based AAMs [96]. This is supported by Duxson et al. in a which Si/Al = 1.9 was found to achieve the optimal strength [97]. While Fletcher et al. [39] reported that NaOH activated metakaolin samples possessed the maximum strength at Si/Al = 8. This can be related to the content difference of reactive Si and Al in the original prime material. Based on extensive investigation on several types of FA, Fernandez-Jimenez et al. [98] concluded that a Si/Al ratio of reactive phase in the range of 1.42–2.38 is suggested. Provis et al. extended the statistic to frequently used precursor systems and summarized that product formed at compositional range of 1<Si/Al < 5 would be suited to general construction applications [18]. Mixtures with superior strength in Figure 2 are with Si/Al ratio locating in this range.

### 4.2. Microstructure

The microstructure of SCAAMs has been intensively characterized using SEM. Specimens produced using GGBS showed a dense and homogeneous microstructure, with little unreacted particles [82]. Voids and cracks can also be observed, which might be related to air bubbles introduced during sample preparation, spaces induced by water evaporation, repaid setting and self-drying of the sample, shrinkage of reaction products, and load application during test [28,99,100]. In comparison, the structure of SCAAMs containing FA is relative loose and porous (Figure 3). Voids, cracks, and unreacted particles can be clearly seen. This can be partially ascribed to the low reactivity of FA, with limited gels to bridge unreacted particles. Moreover, the spherical particle of FA might also be responsible. As depicted by Fernandez-Jimenez et al. [101], dissolution of FA particles initiated from the contact area with activator, leads to exposure of the inner sides or the smaller particles initially entrapped by larger ones. With the diffusion of alkaline solution, chemical attack then takes place in bilateral directions, i.e., from outside in and from inside out. Reaction products are simultaneously formed, covering unreacted parts as a crust and hindering further reaction. 

Reducing the FA content from 70% to 50% and 30%, meant that the numbers of cracks, pore, unreacted particles and unevenness clearly dropped (Figure 3). This affirmed that increasing FA content can result in a high degree of porosity and poor morphology, which might be responsible for the undesired mechanical performance of obtained SCAAMs. Microstructure modification was observed after adding fine particles. Relatively fewer voids and cracks have been detected from samples containing NS or RHA [28,101]. The fine particles can not only fill in voids or cracks but also accelerate formation of reaction products [28]. Patel and Shah stated that the SiO_2_ in nano particles can form siloxo chains (Si-O-Si), firmly reducing the gaps among reaction products and bonding particles. All these are beneficial for generating a dense structure. 

### 4.3. Water Absorption

The water absorption capacity of hardened matrix depends on size distribution and continuity of pores; thus, this characterization can provide an idea about pores present inside. Values of 2.4–3.5% and 2.1–3.4% were obtained by Manjunath et al. for GGBS-based SCAAMS with different mixtures [26,27]. These values are marginally lower than conventional AAM concrete and can be classified as good concrete category [102,103]. Low water absorption suggests the produced SCAAMS are potentially durable and can perform well in aggressive conditions. The water adsorption property can be significantly affected by mixture proportion. The water absorption has been observed to increase from 6.6% to 9.8% when replacing GGBS by FA from 0% to 70% [82]. This is attributed to the porosity increase caused by a boost in unreacted FA particles. Dener et al. [80] studied the influence of aggregate, in which pumice was used as lightweight aggregate. It was reported that the increasing pumice aggregate directly enhanced the water absorption values. The values for mixtures with 60% pumice aggregate were located in the range of 17.96–18.57%, which were increased to 20.71–22.46% when 80% pumice aggregate were used.

The water absorption rate has been investigated by weighing the absorbed water quantity in a time scale of up to 360 min [28]. The values were at a range of 0.069–0.136 mm/√min, with the maximum value being observed for FA-based samples. Specimens containing GGBS generally achieved lower absorption rate and addition can further modify the absorption behavior. Mixing 5% RHA with GGBS contributed to a reduce in the absorption rate to 0.069 mm/√min. This once again supports the above statements about the structure-refining effect of fine additives. 

Practices were developed by plotting water absorption values with corresponding compressive strength, and the results are illustrated in Figure 4. Water absorption decreases with an increase of compressive strength because of denser microstructure and reduced porosity. A close to linear relationship can be inferred from these data, which is similar to OPC concrete and conventional AAMs [104]. 

### 4.4. Acid and Sulfate Resistance

As construction materials, SCAAMs might be used at field sites with varying geochemical parameters. Characterizing the durability over acidic and sulfate condition is fundamental in tailoring potential field applications. The corresponding mass and strength loss has been reported as 2.86–7.74% and 8.76–24.83% [81], for which the acid concentration was 3% (pH = 3) and immersion duration was 90 days. Similar values have been reported under comparable testing conditions but using specimens of varied activator concentrations and adoptions [88]. In comparison with OPC concretes, SCAAMs exhibited better acidic and sulfate resistance with lower mass and strength loss. In the OPC system, degradation of Ca(OH)_2_ forming expansive CaSO_4_ leads to softening and scaling, causing severe deterioration of the matrix [105]. In contrast, the cross-linked aluminosilicate gels of AAMs possess superior resistance to acid attack. Deterioration extent also depends on exposure conditions. By immersing GGBS-based SCAAM in 10% H_2_SO_4_ solution for 12 months, weight and strength loss of 2.2% and 74% were detected, respectively [9]. Thus, increasing acid concentration and exposure duration would result in more severe damage.

Optimizing mixture composition can effectively enhance the resistance of obtained SCAAMs. The influence of precursor and activator have been investigated [81,88]. Increasing NaOH concentration from 2 mol/L to 12 mol/L can obviously reduce the mass and strength loss, while changes in the NaOH/sodium silicate ratio behaved on the contrary. In the studied range of 2–4.5, a higher NaOH/sodium silicate ratio caused more considerable mass and strength loss. In terms of binder composition, the strength loss ratio of FA-based samples was about two times that of the GGBS-based samples, while the mass loss behaved in an opposite way. An interesting study conducted by Huseien et al. [9] illustrated that the addition of CP can improve the acid resistance of GGBS-based SCAAM (Figure 5). Both the mass and strength loss can be gradually reduced with the increase of CP from 0–80%. In comparison with the control sample, mass and strength loss of mixture with 80% CP was decreased from 2.2% to 0.39% and from 74% to 13.3%, respectively. The enhancement in acid resistance was result of the reduction in the free calcium component [9]. All of these bring new insights into the mix-design of SCAAMs with excellent acid resistance.

Similar to the trend observed in acidic conditions is the mass and strength decrease after immersion in sulfate solution. The decrease scale was determined by mixture composition, sulfate concentration and exposure period. A strength drop of 3.6–4.2%, 8.4–9.3%, and 5.6–6.4% was found for GGBS, FA and GGBS-FA based SCAAMs, respectively [81,88]. The corresponding mass reduction can be 1.8–2.1%, 3.5–3.7%, 2.4–2.9%. The deterioration was mainly induced by formation of calcium sulfate dehydrate, which caused expansion and cracking of the matrix. A sample with low, or even free of, Ca(OH)_2_ normally showed good sulfate resistance. This is the case of better resistance of AAMs than OPC concretes. In addition, a compact structure is essential for sulfate resistance, since the inner pore structure provides diffusion pathway for sulfate ions. 

## 5. Other Properties

### 5.1. Impact Resistance

Impact resistance has been characterized by a drop-weight test following ACI Committee 544 recommendations [101]. During the test, a drop hammer from a certain height is periodically impacted to the middle of the specimen. Numbers to cause failure were documented and used to calculate impact energy according to
E_imp_ = N × m × g × h
where E_imp_, impact energy, in J; N, drop numbers; m, mass of drop hammer, in kg; g, gravitational acceleration, 9.81 m/s^2^; h, releasing height of drop hammer, in cm.

The GGBS-based SCAAM achieved adequate impact energy comparing with OPC concrete [106]. Incorporating steel fiber can significantly enhance the impact resistance (Figure 6). Impact energy of samples with 0.5% and 1% short steel fiber (Keremix 30/40) were 5 and 20.5 times of the control. The enhancement was further improved by using long steel fiber (Dramix 60/80). This can be related to the higher friction/adherence of long steel fiber with matrix. Similar phenomena were observed from OPC concrete system [107]. When nanosilica was put into the mixture, the impact energy for mixtures with and without steel fibers can be further enhanced by 1.3–3.5 times [101]. The enhancement was attributed to the improvement in microstructure, especially reduced porosity and modified interfacial transition zone, which affect fracture propagation directly [108].

### 5.2. Interface Shear Strength

Interfacial shear bond strength of SCAAMs has been characterized and compared with self-compacting OPC concrete [109]. Both conventional and modified push-off test (push-off test with normal load over shear interface) were considered. Schematic diagrams and photos of the tests are shown in Figure 7, the details about experimental procedure are referred to [109] and will not be repeated herein. 

After the conventional push-off test, failure was observed in weak zones because of interface shear and slip (Figure 7). In particular, spalling can be seen at the jointing of old and new concretes, which is the region of stress concentration [109]. The ultimate load and corresponding slip were plotted and illustrated as Figure 8, with the highest and lowest interfacial shear stress being achieved by SCAAM-SCAAM samples cast monolithically and SCC-SCAAM specimens produced under cold joint condition, respectively. As stated, stress was directly taken by concrete covering reinforcement and the debonding of concrete to steel resulted in sudden failure of monolithically casted samples. In contrast, the substrate of specimens produced under cold joint condition or laid over hardened concrete can provide enough strength to clamp reinforcement and to stay intact with surrounding concretes. Failure in this case is experienced in stress affecting reinforcement, which causes a well-defined yield plateau as shown in Figure 8 [109]. A similar trend can be inferred in the push-off test scenario of normal load applied over shear interface, in which SCAAM cold joint specimens exhibited higher interface stress than counterparts. All the results demonstrated that SCAAMs possessed better properties than SCC and can be used in concrete repair/rehabilitation.

## 6. Challenges and Perspectives

SCAAMs possess excellent engineering properties and are finding potential applications in civil construction sites. Considering that the manufacture and characterization of SCAAMs are still in laboratory, challenges in mixture design and durability analysis should be addressed to promote the understanding and application of SCAAMs.

Mixture design is a vital procedure for production of concrete materials. The current fabrication of SCAAMs is mainly based on the so called “empirical design method”, in which empirical data of the water-binder ratio, aggregate content, etc., are used for decision of initial mix proportions, and satisfactory mixture for required properties is then concluded after several trial mixes and adjustment. This kind of procedure is easy to follow, however intensive work is needed to obtain an optimal mixture proportion. And any change in source materials or application situation requires intensive re-testing and adjustments. For SCC, research has been conducted and several categories of design methods have been drawn, such as the compressive strength method, close aggregate packing method, statistical factorial model method, rheology of paste model method [1]. These methods possess advantages including precisely determining quantity of specific ingredient, minimizing trial batches, simplifying test protocol, providing basis for quality control, etc., that can be considered as reference for SCAAMs design in the future.

Though several investigations have been conducted on the durability of SCAAMs, this kind of work is still limited in water absorption, acid and sulfate resistance, and data are interpretated in qualitative scale. The durability depends strongly on the nano-properties and microstructure of reaction products and their interaction with surrounding environment. For reliable evaluation of durability and prediction of service life of SCAAMs, characterization under real-world service conditions with multiple modes of corrosion, figuring out key degradation mechanisms, understanding the interrelationships between microstructure and durability, are the key areas with need for future work.

## 7. Conclusions

Recent progress in SCAAMs are revisited and perspectives for further development are discussed in this work. Both fresh and hardened properties of SCAAMs are determined by physicochemical characteristics of source materials and mixture formulation. Though located in a similar range to conventional AAMs, the SCAAMs possess much better workability because of a proportional binder design, high paste-to-aggregate ratio, reduced coarse aggregate content. Benefiting from the accelerated dissolution of aluminosilicate components and formation of strength-giving products, increasing Na_2_Oeq content in binder can generally lead to an improvement in compressive strength. Adoption of fine additives like RHA and NS contributes to enhance mechanical properties and microstructure of obtained SCAAMs, reflecting as reduced water adsorption rate. Based on the general water adsorption values, SCAAMs can be classified as a good concrete category. The SCAAMs exhibited well acidic and sulfate resistance with low mass and strength loss. Optimization in mixture composition, such as increasing activator concentration, incorporating fine particles can further improve the acid and sulfate resistance of SCAAMs. SCAAMs also possess adequate impact energy, which can be considerable enhanced by incorporating steel fiber. The enhancement can be individually further improved by using long fiber or adding nanosilica, because of the high friction/adherence of long steel fiber with matrix and improvement in microstructure. In addition, excellent repair/rehabilitation property of SCAAMs was detected using both conventional and modified push-off tests. All these indicate that SCAAMs can be used as advanced construction materials for civil infrastructure.

Meanwhile, current production and characterization of SCAAMs are still in laboratory scale, and the decision for the initial mixture proportion was frequently based on empirical design method, requiring intensive work and adjustment to obtain optimal proportion. Advanced mix-design procedures should and need to be developed. Moreover, durability characterization of SCAAMs under real-world service condition with multiple modes of corrosion, figuring out key degradation mechanisms, understanding the interrelationships with microstructure, are required for reliable evaluation of durability and prediction of service life. Further work on these areas would contribute to the promotion of understanding and application of SCAAMs.

## Figures and Tables

**Figure 1 molecules-27-00081-f001:**
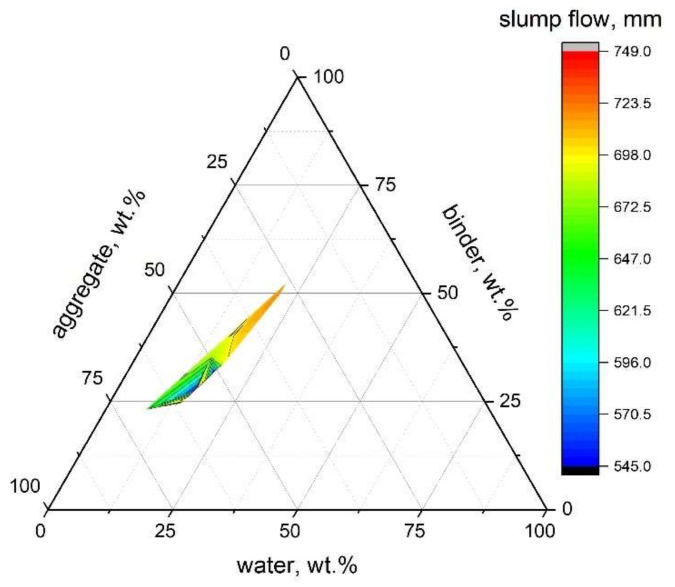
Relationship between slump flow and mixture proportion (data from [9,27,30,81,82,84,85]).

**Figure 2 molecules-27-00081-f002:**
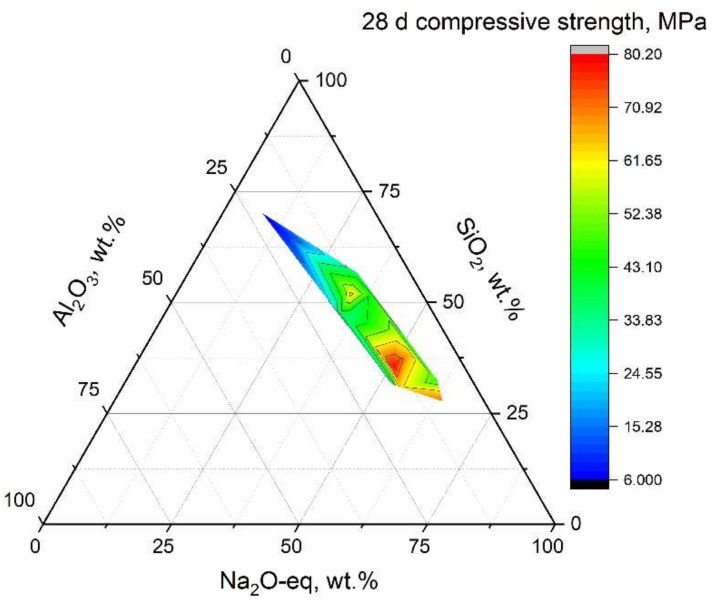
Pseudo-ternary plot of 28 d compressive strength with binder composition (data from [9,27,30,81,82,94]).

**Figure 3 molecules-27-00081-f003:**
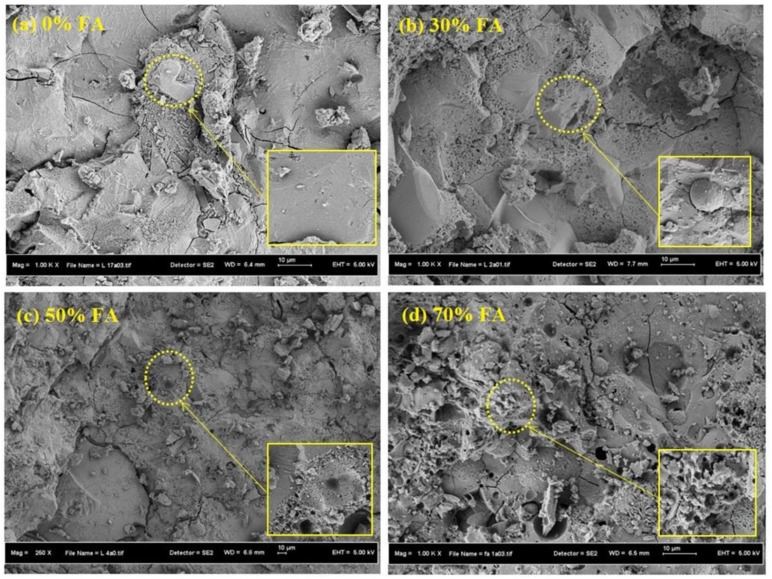
Microstructure of GGBS-based SCAAMs with (**a**) 0%, (**b**) 30%, (**c**) 50% and (**d**) 70% FA [82].

**Figure 4 molecules-27-00081-f004:**
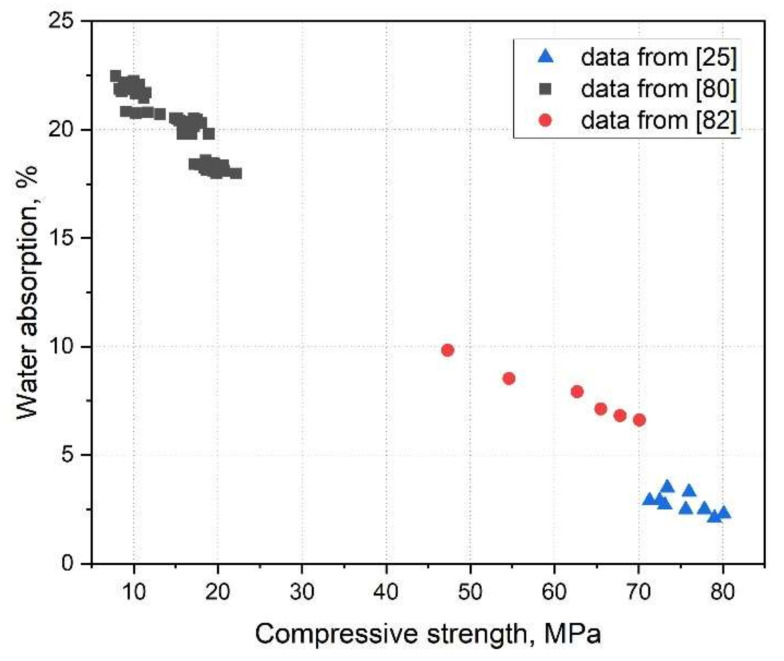
Correlations of water absorption with compressive strength (data from [27,80,82]).

**Figure 5 molecules-27-00081-f005:**
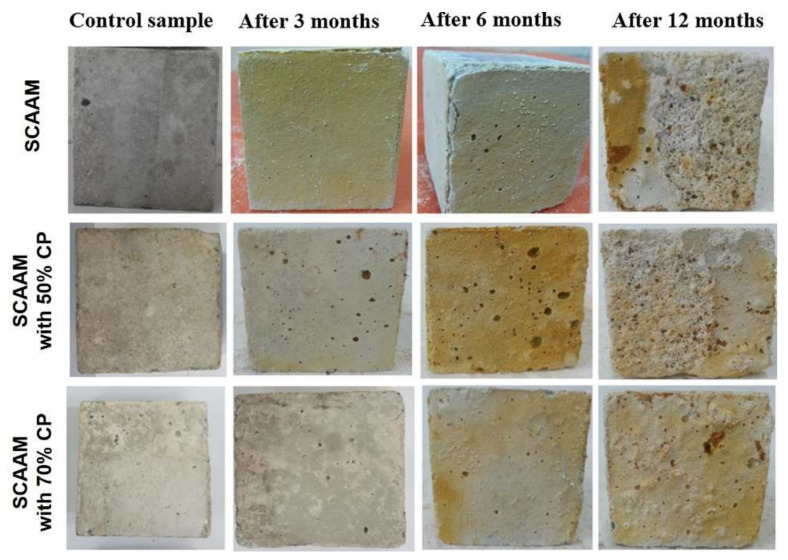
Photos of SCAAMs after acidic exposure (adapted from [9]).

**Figure 6 molecules-27-00081-f006:**
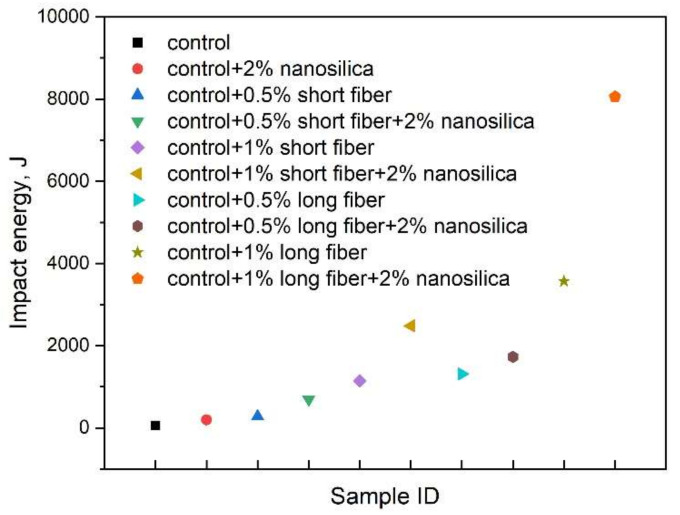
Impact energy of SCAAM samples (reproduced from [101]).

**Figure 7 molecules-27-00081-f007:**
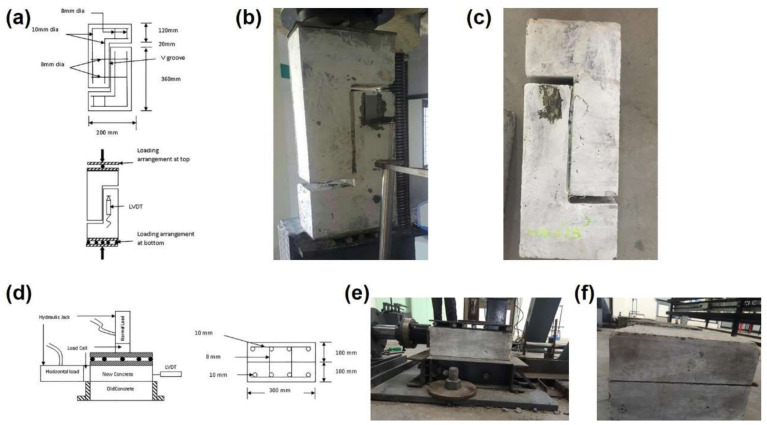
Schematic diagrams and photos of push-off test ((**a**–**c**): conventional push-off test; (**d**–**f**): modified push-off test. (**b**) and (**e**) before test, (**c**) and (**f**) after test) [109].

**Figure 8 molecules-27-00081-f008:**
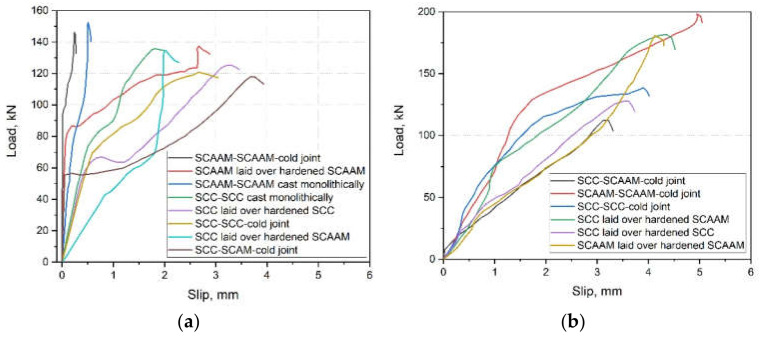
Correlation of load and slip in conventional (**a**) and modified (**b**) push-off test [109].

**Table 1 molecules-27-00081-t001:** Typical SCAAM mixture proportions and properties.

Reference	Precursor	Activator	Fine Aggregate	Coarse Aggregate	Slump Flow		V-Funnel	L-Box	J-Ring	
					Spreading (mm)	T50(s)	(s)	H2/H1	(mm)	Difference
[9]	Mixture of GGBS and ceramic tile powder, 484 kg/m^3^	Mixture of NaOH and sodium silicate solution, activator to solid precursor ratio of 0.5	River sand, 844 kg/m^3^	Crushed granite, 756 kg/m^3^	560–748.4	3.0–6.0	7.5–14.0	0.78–0.95	-	6.0–12.0
[26]	Mixture of GGBS with quartz powder, 700–800 kg/m^3^	Mixtures of sodium silicate with NaOH, solution to precursor ratio 0.4–0.44	Mixture of steel slag sand and quartz sand, 573–728 kg/m^3^	Electric arc furnace slag, 387–490 kg/m^3^	700–800	4–5	8.5–10.4	0.88–0.95	693–795	7.8–9.7
[27]	GGBS, 700–900 kg/m^3^	Mixtures of NaOH and sodium silicate solutions, water to binder ratio of 0.47, 0.475 and 0.48	Slag sand, 303–636 kg/m^3^	Electric arc furnace slag, 228–480 kg/m^3^	685–720	3.7–4.8	8.5–11.0	0.86–0.96	670–717	5.0–7.0
[30]	Mixture of GGBS and FA, 450 kg/m^3^	Sodium silicate and sodium hydroxide solutions, activator to precursor 0.5	Crushed limestone, 859.7–865.6 kg/m^3^	Crushed limestone, 737.8–742.8 kg/m^3^	679.5–709.0	2.8–3.6	10.8–16.6	0.88–0.96	-	-
[80]	Mixture of GGBS with RHA, 500 kg/m^3^	Mixture of sodium silicate solution with NaOH	River sand, 1100 kg/m^3^	Crushed limestone, 785 kg/m^3^	655–710	4.0–5.5	8–13	0.8–0.95	-	6–9
[81]	Mixture of FA and GGBS, 480–655 kg/m^3^	Mixture of NaOH and sodium silicate solution, activator to precursor ratio of 0.1, 0.3 and 0.5	-	-	545.7–706.5	3.0–14.03	9.0–19.0	0.38–0.88	-	-
[82]	Mixture of GGBS and FA, 484 kg/m^3^	Solution with SiO_2_-to-Na_2_O ratio of 1.02, activator to precursor ratio of 0.5	River sand, 844 kg/m^3^	Crushed limestone, 756 kg/m^3^	560.6–720.2	3.5–6.0	8.5–14.0	0.78–0.92	-	6.0–12.0
[83]	GGBS, 475 kg/m^3^	Mixture of NaOH with sodium metasilicate, activator to precursor ratio of 0.43	Mixture of spent garnet with sand, 950 kg/m^3^	-	671–700	3.5–5.5	6.5–12.0	0.91–0.97	-	-

## Data Availability

Not applicable.

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
