# Peer review of "Self-Compacting Alkali-Activated Materials: Progress and Perspectives"

_molecules, 2021, doi:10.3390/molecules27010081_

Round 1

Reviewer 1 Report

As authors are mitigated all comments, therefore I would like to recommend it for publication in your journal.

Author Response

Many thanks for the comment, the manuscript has been revised.

Reviewer 2 Report

This paper revisits the recent progresses in SCAAMs and perspectives for further development. Thus, it is assumed as a review paper. However, this paper provides direct comparison and interpretation of data from different publications and shows the normalization of the mixture in terms of the content of key reactive components from precursors and activators, correlation with mechanical behaviors. Authors must have knowledge of the technical writing of the review paper. The paper is too ambiguous. There is too much detail of literature in this paper whereas the paper title doesn’t indicate that it is a review paper because a review paper must have authors critical observations/opinions on the published results and should also indicate or identify future research directions, which is missing.

Section 2 Test and basic requirement for SCC is unnecessary and well-established therefore can be omitted. Table 1 is enough as a summary of Section 2.

Table 2 is too long and complex as it summarises the typical SCAAM mixtures from the literature.

In Section 5. Property of fresh SCAAMs, recalculation of masses was conducted to facilitate and simplify the comparison of data from different sources. Why? However, this section should have shown the results of fresh SCAAMs.

Author Response

We thank the reviewer very much for reviewing the manuscript. Revision has been made by re-organizing the structure (section 2 in last version was depleted, Table 2 was optimized).

SCAAMs are produced using different precursors with different physicochemical characteristics and different proportions, comparison can hardly be made. In this work, re-calculation was made with the purpose to figure out the key parameters dominating the properties of SCAAMs. 

Reviewer 3 Report

This paper presents review study of recent progresses in self-compacting alkali-activated materials, including mixture proportions, fresh properties, mechanical strength, microstructure, acid and sulfate resistance, high temperature behaviors, impact resistance and interface shear strength. Authors performed direct comparison and interpretation of data taken from different papers. What is more they normalized mixture proportions contents in terms of checking the corelations. 

Although presented article may be interesting to the readership of this journal, the paper may only be considered for publication after the following concerns have been addressed successfully in a revision:

In general:
1) I suggest the authors rethink the number of subsections. In my opinion, there are too many of them and they could be grouped into overarching chapters, e.g. materials, properties, application and methods of analysis etc. Moreover, starting with "2. Test and basic requirement for SCC" is probably not the best idea, as the authors start discussing tests without introducing the materials. I think this paragraph should be more informative as the begining of the Review.

Regarding to the "3. New precursors" part:
2) I recommend authors to add some add tabular comparison for new precursons as well. The description in its current form is overwhelming with the text, from which it is difficult, however, to draw conclusions without searching in the references.  

Regarding to the "4. Production of SCAAMs" part:
3) Presentation of Table 2 could be better. As it stands, data is difficult to read. Perhaps it would be better to swap rows and columns to avoid stretching the entire table. Moreover, there is no reference to Table 2 in the text (Maybe due to lots of Reference errors (?). Please compare with "Minor errors" part). 

Minor errors:
4) Lines 74-75, 89, 98, 111, 125-126, 222, 245-246, 279, 300-301, 344-345, 384, 393, 405, 440, 467, 501, 516, 522, 524-525, 533: Some reference errors are present. Please check it.
5) Line 140: For "Na2O" subscript should be used for the digit 2.

Author Response

We thank the reviewer very much for the critical comments and suggestions. Revisions have been made following the comments. The manuscript structure has been re-organized. Section 2 "Test and basic requirements" has been depleted. Section 3 "New precursors" and Section 4 "Production of SCAAMs" have been merged into "Materials and formulations". Table 2 has also been optimized to make it clear and easy to read. Minor errors have been revised.

Round 2

Reviewer 2 Report

The authors have significantly improved the manuscript. However, technical detail is still missing in this manuscript in spite of reviewing 109 research papers. Following are the comments.

In this whole manuscript, the authors' technical comment or view in each result section is required as an essential constituent of a review paper. Only summarizing the data from various papers is not a review paper.

The discussion on the results should be in more detail.

Microstructure images are provided only for Fly Ash but not for other SCMs.

It is suggested to include the compressive strengths achieved by the researchers in Table 1 to make it more explanatory

In section 4.1 heading, the title should specifically mention, which strength authors are referring to?

It was earlier suggested to use the specific symbol “×” for multiplication. The symbol “*” should be avoided wherever applicable

Please include the full form of C-A-S-H and C-S-H at least once. This comment is applicable to all shorthand forms used.

Author Response

The authors have significantly improved the manuscript. However, technical detail is still missing in this manuscript in spite of reviewing 109 research papers. Following are the comments.

In this whole manuscript, the authors' technical comment or view in each result section is required as an essential constituent of a review paper. Only summarizing the data from various papers is not a review paper.

The discussion on the results should be in more detail.

We thank the reviewer for the comments. This manuscript is however not “Only summarizing the data from various papers”, discussions were made everywhere throughout the manuscript. Moreover, we conducted extensive re-calculation of data from various papers and discussions to reveal the key parameter influencing the physicochemical properties of SCAAM. We respect the reviewer’s time and efforts in reviewing this work, and we also ask the reviewer to respect our efforts.

Microstructure images are provided only for Fly Ash but not for other SCMs.

Since comparisons and discussions were given in the manuscript, representative microstructure images were then presented. We believe there’s no necessary to provide all related images in a manuscript.

It is suggested to include the compressive strengths achieved by the researchers in Table 1 to make it more explanatory

Table 1 is designed to show the influence of mixture on workability properties of SCAAMs, strength is then the point of section 4. To make the manuscript concise, strength data were not given in Table 1.

In section 4.1 heading, the title should specifically mention, which strength authors are referring to?

This has been revised as commented.

It was earlier suggested to use the specific symbol “×” for multiplication. The symbol “*” should be avoided wherever applicable

We checked the whole manuscript, this has been revised following the comment.

Please include the full form of C-A-S-H and C-S-H at least once. This comment is applicable to all shorthand forms used.

Thanks for the comment, this has been revised.